# Context Matters: Anatomy-Aware Dual-Stream Multiple Instance Learning Framework for eGFR Prediction

**Abteen Arab**                                      AARAB02@STUDENT.UBC.CA
**Maziar Riazy**[*]                                  MAZIAR.RIAZY@PHC.CA
**Ali Bashashati**[*]                                ALI.BASHASHATI@UBC.CA
*University of British Columbia, Vancouver, Canada*

## Abstract

Current deep learning approaches for modeling kidney disease typically rely on pathologists' annotations, which are costly to obtain and subject to inter-observer variability. In contrast, estimated glomerular filtration rate (eGFR) is a routine clinical measure of kidney function that offers a more scalable supervision signal, yet remains relatively underexplored. In this work, we assess the performance of multiple instance learning (MIL) frameworks for eGFR prediction from kidney whole slide images (WSIs). Importantly, we show that the predictive value of different tissue segments is disease dependent and that modeling kidney disease as two anatomy-defined MIL streams yields stronger performance than standard single-stream baselines.

**Keywords:** Kidney Disease, Computational Pathology, Multiple Instance Learning (MIL)

## 1. Introduction

Pathological assessment of kidney disease using deep learning has generally followed two supervision paradigms: histopathology based modeling and function guided modeling. Histopathology based approaches use expert annotations as supervision, but these labels are costly to obtain and are subject to interobserver variability (Boud'hors et al., 2022; Weis et al., 2022; Kurata et al., 2025). In contrast, function guided modeling uses clinical measures of renal function, most commonly eGFR, as supervision (Cho et al., 2025; Kasireddy et al., 2026). In this setting, the goal is not merely to predict eGFR, but to learn WSI representations that capture function-related morphologic signals. Compared to morphology-based labels, eGFR is more readily available while remaining strongly associated with kidney function and outcome.

Prior work on eGFR prediction from WSIs has mainly followed two formulations: patch-level binary classification and whole-slide MIL regression. Earlier studies typically use selected regions of interest to predict coarse eGFR categories (Cho et al., 2025), whereas Kasireddy et al. (2026) consider the task in a whole-slide MIL setting with a regression head. These two settings offer complementary advantages and limitations. Patch level methods can focus on highly informative local regions, but may miss broader tissue context. Whole slide approaches preserve broader context, but may dilute region specific predictive signals such as kidney anatomy. This suggests the need for a framework that retains slide level context while explicitly modeling relevant tissue regions.

---

[*] Co-Corresponding Authors

To address this, we propose an anatomy-aware dual-stream MIL framework for baseline eGFR prediction from WSIs. Rather than representing each slide as a single homogeneous bag, our approach decomposes the WSI into glomerular enriched and non-glomerular patches, processes them as separate MIL streams, and integrates them through a learned bidirectional gating mechanism. Our contributions are threefold: (1) we formulate eGFR prediction from WSIs as an anatomy-aware MIL problem; (2) we show that anatomical compartments carry distinct eGFR predictive signals, and that their predictive value differs depending on disease phenotype; and (3) we propose a dual-stream gating strategy for integrating compartment specific signals into a slide level representation.

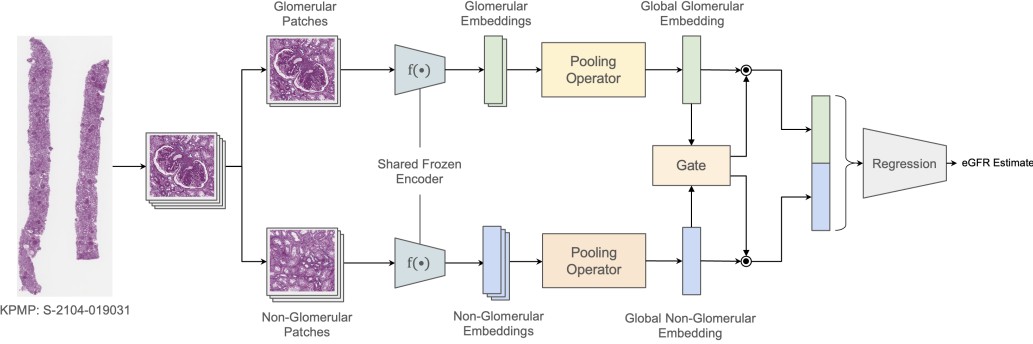

Figure 1: Overview of the anatomy-aware dual-stream MIL architecture.

## 2. Dataset, Models, & Methods

**Pooling Framework.** As shown in Figure 1, each WSI is represented by two bags, glomerular $\mathcal{G}$ and non-glomerular $\mathcal{N}$, which are embedded using a shared frozen CONCHv1.5 encoder $f(\cdot)$ (Lu et al., 2024). Pooling operators are applied to each bag to obtain global glomerular and non-glomerular representations, $\mathbf{h}_G$ and $\mathbf{h}_N$, respectively. We then apply bidirectional feature wise gating, where gating vectors $\mathbf{r}_G = \sigma(\mathrm{MLP}_G([\mathbf{h}_G; \mathbf{h}_N]))$ and $\mathbf{r}_N = \sigma(\mathrm{MLP}_N([\mathbf{h}_G; \mathbf{h}_N]))$ are used to modulate the global compartment representations via element wise multiplication, yielding $\tilde{\mathbf{h}}_G = \mathbf{r}_G \odot \mathbf{h}_G$ and $\tilde{\mathbf{h}}_N = \mathbf{r}_N \odot \mathbf{h}_N$. The resulting representation $[\tilde{\mathbf{h}}_G; \tilde{\mathbf{h}}_N]$ are passed to a regression head to estimate eGFR.

**Datasets.** We train and evaluate all models on 696 PAS-stained WSIs from 347 patients in the Kidney Precision Medicine Project (2026). Full details on the dataset, pre-processing steps, and model can be found in Appendix A & B.

**Experiments & Evaluation.** We compare our anatomy-aware framework against standard MIL baselines, including mean pooling, ABMIL (Ilse et al., 2018), and TransMIL (Shao et al., 2021), in two settings: *single-stream* and *dual-stream*, which differ in the number of MIL streams used. Within these settings, models take as input either the whole slide, glomerular enriched patches, or non-glomerular patches. For dual-stream models, we also evaluate two ablations: unidirectional gating and post-pooling late fusion. All models were evaluated using 3-fold patient-level cross-validation across 3 seeds, and performance was reported fold-wise using root mean square error (RMSE) alongside regression metrics.

## 3. Results & Discussion

Table 1: eGFR regression performance. **Bold** indicates the best model, underline indicates second best model, and double underline indicates third best model.

| Representation | Stream Setting | Pooling Operator | Pearson's $r \uparrow$ | Spearman's $\rho \uparrow$ | RMSE $\downarrow$ |
|---|---|---|---|---|---|
| Dual-Stream | Dual Gate | TransMIL | $0.5048_{\pm 0.0235}$ | $0.4684_{\pm 0.0413}$ | $22.53_{\pm 3.4903}$ |
| | | **ABMIL** | $\mathbf{0.6236}_{\pm 0.0506}$ | $\mathbf{0.5497}_{\pm 0.0748}$ | $\mathbf{19.07}_{\pm 3.3059}$ |
| | | Mean Pool | $0.5248$$_{\pm 0.0477}$ | $0.4893$$_{\pm 0.0793}$ | $22.14_{\pm 2.8999}$ |
| | Glomerular Gate | TransMIL | $0.4612_{\pm 0.0410}$ | $0.4165_{\pm 0.0378}$ | $23.24_{\pm 3.6077}$ |
| | | ABMIL | $0.4978_{\pm 0.0280}$ | $0.4521_{\pm 0.0452}$ | $22.31_{\pm 2.6821}$ |
| | | Mean Pool | $0.5083_{\pm 0.1149}$ | $0.4839_{\pm 0.0975}$ | $22.30_{\pm 4.3130}$ |
| | Non-Glomerular Gate | TransMIL | $0.4964_{\pm 0.0187}$ | $0.4815_{\pm 0.0328}$ | $23.28_{\pm 2.2006}$ |
| | | ABMIL | $0.4716_{\pm 0.0421}$ | $0.4493_{\pm 0.0525}$ | $22.88_{\pm 2.6494}$ |
| | | Mean Pool | $0.4624_{\pm 0.0281}$ | $0.4174_{\pm 0.0430}$ | $22.97_{\pm 2.5062}$ |
| | Late Fusion | TransMIL | $0.4939_{\pm 0.0271}$ | $0.4498_{\pm 0.0663}$ | $22.99_{\pm 3.0887}$ |
| | | ABMIL | $0.4938_{\pm 0.0088}$ | $0.4767_{\pm 0.0115}$ | $23.65_{\pm 2.8058}$ |
| | | Mean Pool | $0.4669_{\pm 0.0682}$ | $0.4321_{\pm 0.0841}$ | $22.58_{\pm 3.3420}$ |
| Single-Stream | All Patches | TransMIL | $0.4734_{\pm 0.0073}$ | $0.4311_{\pm 0.0108}$ | $23.81_{\pm 2.3466}$ |
| | | ABMIL | $0.5214_{\pm 0.0400}$ | $0.4850_{\pm 0.0434}$ | $21.90$$_{\pm 3.0088}$ |
| | | Mean Pool | $0.4993_{\pm 0.0625}$ | $0.4671_{\pm 0.0781}$ | $22.47_{\pm 3.4559}$ |
| | Glomerulus Enriched Patches | TransMIL | $0.4805_{\pm 0.0585}$ | $0.4337_{\pm 0.0566}$ | $22.50_{\pm 4.1410}$ |
| | | ABMIL | $0.4380_{\pm 0.0729}$ | $0.4001_{\pm 0.0780}$ | $23.30_{\pm 3.5793}$ |
| | | Mean Pool | $0.4874_{\pm 0.0143}$ | $0.4528_{\pm 0.0395}$ | $22.45_{\pm 2.8337}$ |
| | Non-Glomerular Patches | TransMIL | $0.5031_{\pm 0.0456}$ | $0.4729_{\pm 0.0518}$ | $22.51_{\pm 3.0477}$ |
| | | ABMIL | $0.5478$$_{\pm 0.0579}$ | $0.5165$$_{\pm 0.0504}$ | $21.70$$_{\pm 3.1528}$ |
| | | Mean Pool | $0.4802_{\pm 0.0946}$ | $0.4579_{\pm 0.0944}$ | $22.63_{\pm 4.1248}$ |

Table 1 summarizes eGFR regression performance. Among single-stream MIL models, ABMIL applied to the non-glomerular compartment performs best overall and surpasses the whole slide baselines, while glomerulus-enriched models perform similarly to the whole slide models. When stratified by primary disease site, however, glomerulus-enriched models perform best in glomerulocentric diseases, whereas non-glomerular models perform best in extra-glomerular diseases (Appendix C). This suggests that predictive signal is disease dependent and that aggregating an entire slide into a single bag can dilute region specific predictive signal. We therefore ask whether modeling multiple MIL streams can improve prediction. Simple fusion strategies such as late fusion and unidirectional gating do not improve over the strongest single-stream baselines, whereas bidirectional gating slide-level conditioned on the whole slide image performs best ($p = 5 \times 10^{-4}$).

## 4. Conclusion & Future Work

Overall, these findings suggest that renal modeling benefits from decomposing tissue into anatomically meaningful compartments and from using an effective mechanism to integrate compartment-specific MIL streams. An important direction for future work is to test whether these function guided representations transfer to downstream clinical endpoints such as risk of kidney failure or lesion chronicity scoring.

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

## Appendix A. Data Set & Pre-Processing

**Dataset.** This study used the Kidney Precision Medicine Project (2026) cohort which consists of 1,147 WSIs with matched eGFR data. We kept only cases with at least 30 glomerular enriched patches, a commonly used threshold for an adequate renal biopsy (Boud'hors et al., 2022), resulting in a final cohort of 696 WSIs from 347 participants. By primary adjudicated disease, this cohort included diabetic kidney disease ($n = 244$), hypertensive kidney disease ($n = 152$), acute tubular injury ($n = 62$), acute interstitial nephritis ($n = 22$), healthy control ($n = 28$), and cases for which the diagnosis could not be determined ($n = 188$).

**Patch Extraction and Glomerular Identification.** We extracted $512{\times}512$ patches at $20\times$ magnification and applied Reinhard and Vahadane stain normalization. Glomerular regions were identified using frozen segmentation models, including a Mask2Former model released through the KPI 2024 challenge (Cap, 2025; Deng et al., 2025). To evaluate sensitivity of this preprocessing step, we measured the average number of glomerular patches per case across different segmentation models and pixel thresholds (rounded to the nearest whole number), and also report model DICE on a subset of cases with glomerular masks. As shown in Table 2, the number of glomerular patches remains fairly stable across models up to a threshold of 5,000 pixels. At higher thresholds, smaller glomeruli begin to be excluded, followed by larger ones around 50,000 pixels. Overall, these results show that bag cardinality is relatively robust to the choice of segmentation model and threshold, which in turn suggests that the downstream dual-stream MIL outputs are insensitive to these factors.

Table 2: Glomerulus Detection Sensitivity Analysis.

| Segmentation | Dice | >0 | ≥500 | ≥1,000 | ≥5,000 | ≥10,000 | ≥50,000 | ≥100,000 |
|---|---|---|---|---|---|---|---|---|
| Mask2Former | 0.90 | 96 | 89 | 80 | 55 | 50 | 19 | 6 |
| UPerNet | 0.88 | 81 | 75 | 73 | 63 | 55 | 20 | 6 |
| SegFormer | 0.85 | 59 | 54 | 53 | 48 | 43 | 19 | 6 |

## Appendix B. Training Protocol

All models were trained under a shared protocol. Specifically, models were trained for 30 epochs with AdamW, a learning rate of $10^{-3}$, weight decay of $10^{-4}$, and dropout of 0.1 when supported by the architecture. Evaluation was performed using 3 fold patient level cross validation, with results reported as the mean metric across test folds. Each WSI was treated as a separate input, while all WSIs from the same patient were assigned to the same fold. All models were optimized using a composite Huber Pearson loss, where the Pearson term ($r(\hat{\mathbf{y}}, \mathbf{y})$) was scaled to be on a range comparable to the Huber term:

$$\mathcal{L} = \mathcal{L}_{\text{Huber}}(\hat{\mathbf{y}}, \mathbf{y}) - 2\, r(\hat{\mathbf{y}}, \mathbf{y}). \tag{1}$$

## Appendix C. Additional Ablation Studies

**Disease-Specific Ablation.** We stratified cases by primary disease site to test whether the predictive value of each compartment depends on disease phenotype. Diabetic kidney disease ($n = 244$) was treated as a relatively glomerulus involved disease, whereas acute interstitial nephritis ($n = 22$) and acute tubular injury ($n = 62$) were treated as relatively extra glomerular diseases. Within each subgroup, we compared ABMIL models trained on glomerular only (G), non glomerular only (N), whole slide single-stream (WS), and whole slide dual-stream inputs. As shown in Table 3, the most predictive single-stream compartment varies by disease phenotype, broadly aligning with the dominant site of pathology. The alternate compartment remains informative in both settings, while the dual-stream model achieves the strongest Pearson's $r$ and competitive RMSE.

Table 3: Disease-specific ablation. **Bold** indicates the best model and underline indicates the second best model.

| Disease Group | Input | RMSE ↓ | Pearson's $r$ ↑ |
|---|---|---|---|
| Glomerulus-Involved | Dual-stream (WS) | 21.21 | **0.5945** |
| | Single-stream (WS) | 22.74 | 0.4585 |
| | Glomerular Enriched | **20.84** | 0.5689 |
| | Non-glomerular | 22.99 | 0.3501 |
| Extra-Glomerular | Dual-stream (WS) | 19.75 | **0.6366** |
| | Single-stream (WS) | 20.88 | 0.5250 |
| | Glomerular Enriched | 23.07 | 0.3518 |
| | Non-glomerular | **19.35** | 0.6019 |

**Fixed-Cardinality Bag Size Ablation.** We perform a fixed-cardinality ablation to isolate the effect of natural bag size by constraining each model to a total bag size of $K \in 16, 32, 64$ over five random seeds; for dual-stream models, patches are split evenly across streams. We report signed $\Delta$RMSE relative to the single-stream ABMIL whole-slide model. As shown in Table 4, single-compartment models consistently outperform the whole-slide baseline under fixed-cardinality sampling, while the dual-stream whole-slide model performs best at larger bag sizes. This suggests that, although bag cardinality contributes to performance, it does not fully explain the trends observed in the main body.

Table 4: Fixed-cardinality ablation reported as $\Delta$RMSE. **Bold** indicates the best model within the given bag size, and underline indicates the second best.

| MIL Method | $K = 16$ | $K = 32$ | $K = 64$ | Natural Bags |
|---|---|---|---|---|
| Dual-Stream (WS) | -0.4509 | -0.5772 | **-0.5299** | **-2.83** |
| Single-Stream (G) | -0.7073 | -0.3259 | -0.4075 | 1.40 |
| Single-Stream (N) | **-0.7916** | **-0.6594** | -0.5041 | -0.20 |

## Appendix D. Transferability of Learned Slide Representations

We evaluated whether function guided slide representations transfer beyond the primary eGFR regression task by freezing the pre-regression slide embeddings from each trained model and using them as input to lightweight MLP classifiers for downstream albuminuria and proteinuria prediction. Albuminuria is typically a marker of glomerular filtration barrier injury, whereas proteinuria is broader and may arise from glomerular damage, impaired tubular reabsorption, or overflow of low-molecular-weight proteins (Comper et al., 2022). Specifically, we define albuminuria prediction as a binary classification task using 30 mg/g creatinine as the cutoff, with cases below this threshold treated as negative and cases at or above it treated as positive. Similarly, we define proteinuria prediction as a binary classification task using 150 mg/g creatinine as the cutoff, with cases below this threshold treated as negative and cases at or above it treated as positive. Transfer performance was evaluated using AUROC, accuracy, and F1 score.

Table 5: Transfer learning performance on alternate renal pathology classification tasks. **Bold** indicates the best value within each target and metric, and underline indicates the second best.

| Target | Stream Setting | Model | Accuracy ↑ | AUROC ↑ | F1 Score ↑ |
|--------|---------------|-------|-----------|---------|-----------|
| Proteinuria | Whole slide | Single-Stream ABMIL | $0.6980_{\pm 0.0409}$ | $0.7465_{\pm 0.0771}$ | $0.7874_{\pm 0.0345}$ |
| | Glomerular | Single-Stream ABMIL | $0.6984_{\pm 0.0489}$ | $0.7217_{\pm 0.0881}$ | $0.7822_{\pm 0.0425}$ |
| | Non-Glomerular | Single-Stream ABMIL | $\underline{0.7390}_{\pm 0.0477}$ | $\underline{0.8087}_{\pm 0.1232}$ | $\underline{0.8102}_{\pm 0.0354}$ |
| | Dual-Stream | Bidirectional Gate | $\mathbf{0.7438}_{\pm 0.0093}$ | $\mathbf{0.8260}_{\pm 0.1074}$ | $\mathbf{0.8160}_{\pm 0.0025}$ |
| Albuminuria | Whole slide | Single-Stream ABMIL | $0.6650_{\pm 0.0109}$ | $0.6363_{\pm 0.1562}$ | $\mathbf{0.7668}_{\pm 0.0176}$ |
| | Glomerular | Single-Stream ABMIL | $\underline{0.6656}_{\pm 0.0487}$ | $\underline{0.7124}_{\pm 0.0949}$ | $0.7601_{\pm 0.0169}$ |
| | Non-Glomerular | Single-Stream ABMIL | $0.6499_{\pm 0.0378}$ | $0.6465_{\pm 0.1846}$ | $0.7470_{\pm 0.0111}$ |
| | Dual-Stream | Bidirectional Gate | $\mathbf{0.6687}_{\pm 0.0387}$ | $\mathbf{0.7242}_{\pm 0.0748}$ | $\underline{0.7648}_{\pm 0.0345}$ |

The transfer results in Table 5 suggest that the most informative representation is target dependent. Among single-stream models, whole slide embeddings are generally less competitive than the strongest compartment-specific representations, suggesting that whole slide aggregation may dilute region-specific signal. For proteinuria, the strongest single-stream transfer performance comes from the non-glomerular model, consistent with the fact that proteinuria reflects a broad range of tissue injury beyond the glomerulus alone. For albuminuria, the glomerular single-stream model is more competitive than the other single-stream models, consistent with albuminuria being more closely tied to glomerular pathology. Despite these target-specific trends in the single-stream setting, the dual-stream representation achieves the strongest overall performance across both tasks, suggesting that joint modeling captures complementary compartment-specific information that transfers more robustly across related functional endpoints.

## Appendix E. Pathological Analysis

To explore the biological signals captured by the dual-stream model, we performed a preliminary pathologic analysis using a pathology-trained vision-language model (VLM). Patients were first stratified into high- and low-predicted eGFR groups. For each stream, high-attention patches were identified using ABMIL attention scores and then compared between groups. These patches were analyzed with Quilt-LLaVA to obtain a broad characterization of morphologic patterns (Seyfioglu et al., 2024), and the patch-level outputs were subsequently aggregated using a separate language model. The prompt used for this analysis is shown below:

---

**Quilt-LLaVA Prompt for Morphologic Feature Extraction**

```
You are analyzing a kidney PAS patch as a renal pathologist.  Analyze the
patch and extract the following:

   • Inflammation:  absent/present/uncertain

   • Fibrosis or matrix expansion:  absent/present/uncertain

   • Tissue injury or degenerative change:  absent/present/uncertain

   • Structural distortion:  absent/present/uncertain

First, provide a summary of the tissue section, and then extract the above
features with supporting comments.  Only comment based on the image.
```

---

Table 6: Percentages of VLM-identified morphologic features across eGFR strata. **Bold** indicates significant difference.

| Feature | Glomerular Patches | | Non-Glomerular Patches | |
|---|---|---|---|---|
| | **High** | **Low** | **High** | **Low** |
| Inflammation | $4.2_{\pm 0.9}\%$ | $\mathbf{14.0_{\pm 1.9}}\%$ | $5.1_{\pm 3.0}\%$ | $3.4_{\pm 5.2}\%$ |
| Fibrosis or matrix expansion | $1.3_{\pm 2.4}\%$ | $\mathbf{6.9_{\pm 0.7}}\%$ | $3.4_{\pm 5.9}\%$ | $2.8_{\pm 0.7}\%$ |
| Tissue injury or degenerative change | $1.8_{\pm 3.1}\%$ | $\mathbf{6.0_{\pm 1.0}}\%$ | $\mathbf{9.9_{\pm 1.1}}\%$ | $2.3_{\pm 1.6}\%$ |
| Structural distortion | $2.5_{\pm 0.3}\%$ | $\mathbf{5.6_{\pm 1.7}}\%$ | $0.9_{\pm 2.2}\%$ | $0.1_{\pm 1.0}\%$ |

The aggregated morphological outputs are summarized in Table 6, with uncertainty estimated via patch-level bootstrapping. The attention-enriched morphologic patterns differ across compartments. In glomerular patches, low eGFR is associated with greater inflammation, fibrosis or matrix expansion, tissue injury, and structural distortion, consistent with more advanced glomerular damage. In contrast, in non-glomerular patches, high eGFR shows greater tissue injury or degenerative change, suggesting that tubulointerstitial abnormalities may remain informative even when filtration is relatively preserved. Together, these findings support the idea that glomerular and non-glomerular regions encode partially distinct signals, and that their relevance to eGFR may depend on disease context and the balance between acute and chronic injury. Further analysis is needed, as these VLM-derived summaries are preliminary and require expert renal pathology review.

