# OpenReview forum: "Context Matters: Anatomy-Aware Dual-Stream Multiple Instance Learning Framework for eGFR Prediction"
_MIDL.io/2026/Short_Papers — MIDL 2026 - Short Papers Poster_

### Official Review · Reviewer_CVMB · 2026-05-04
**Good Idea and Validation, but terminology may be slightly misleading**

**Rating:** 5
**Confidence:** 4

**Review:**

I give a strong accept rating to this poster based on the interesting idea and sufficient experimentation/ablation.

**Summary:**

This paper proposes a method for eGFR prediction from Whole Slide Imaging (WSI). The authors hypothesize that the tissue sections containing glomerules and non-glomerules carry different levels of information and thus need to be processed differently. This paper proposes a two stream approach to process the two different levels of information and a dual stream gating strategy to combine the two different levels of information. The two types of compartments are prepared using a frozen segmentation model including Mask2Former.

**Strengths:**

The idea of extracting anatomy awareness in the form of presence/absence of glomerules in the WSI is interesting.
The authors have performed sufficient validation and ablation to show the effectiveness of their proposed approach.
The dual gating strategy is interesting and is shown to be effective via the different ablation studies performed.

**Weaknesses:**

The naming of ‘anatomy-aware’ might be slightly confusing, since the anatomical awareness is mainly based on the presence or absence of glomerules, which is a relatively narrow aspect of anatomy. In general anatomy-awareness implies a richer understanding such as spatial relationships between structures or morphology etc.

**Justification Of Rating:**

The compartmentalization based on the presence of glomerules is interesting, and sufficient validation is the main reason behind the rating.

---

### Decision · Program_Chairs · 2026-05-08

Accept (Poster)